# The Relationship between Social Support Correlates and Feelings of Loneliness among Male UK Recreational Anglers

**DOI:** 10.3390/ijerph20115997

**Published:** 2023-05-30

**Authors:** Mike Trott, Mark Tully, Andy Torrance, Lee Smith

**Affiliations:** 1Centre for Public Health, Queen’s University Belfast, Belfast BT12 6BA, UK; 2School of Medicine, Ulster University, Belfast BT15 1AP, UK; 3Angling Direct, Norwich NR136LH, UK; 4Centre for Health, Performance and Wellbeing, Anglia Ruskin University, Cambridge CB1 1PT, UK

**Keywords:** angling, fishing, loneliness, friend, relative

## Abstract

The benefits of access to blue spaces (exposure to aquatic environments) have been well reported. One common activity conducted in these spaces is recreational angling. Studies have shown that several correlates are associated with recreational angling, including a lower incidence of anxiety disorders compared to non-anglers. What is currently unknown is how measures of social support relate to feelings of loneliness in this population. The aim of this study, therefore, is to examine experiences of loneliness and social support in male UK anglers. In total, 1752 participants completed an online survey. The results of this study showed that the higher the number of friends or family that anglers hear from and feel close to, the less likely they are to report a lack of companionship, the less likely they are to report feeling left out, and the less likely they are to report feelings of isolation. Furthermore, more than half of the sample reported hardly ever or never having feelings of loneliness, suggesting that recreational angling does not affect feelings of loneliness.

## 1. Introduction

The benefits of access to blue spaces (exposure to aquatic environments such as rivers, lakes, the sea, and canals) are well reported. For example, one systematic review including 35 studies reported positive associations between exposure to blue spaces and mental health, overall wellbeing, and levels of physical activity [1]. The review highlights that these positive associations are likely owing to the fact that visiting blue space reduces psychological stress, and increases social contacts and place attachment. One common leisure activity undertaken in blue spaces is recreational angling, with a reported 105,000 people in England participating in the sport in 2020–2021 [2]. Furthermore, the UK government has reported that recreational angling adds GBP 1.4b into the UK economy [3]. Further, research has shown that there is a lower incidence of anxiety disorders in recreational anglers [4]. Another interesting finding from this study was that anglers were significantly older than non-anglers—indeed, almost half of the population was over 55 years of age [4]. In this older population, several social correlates are decreased compared to younger adults. For example, in a large cohort study with a 28-year follow-up, high levels of loneliness (i.e., the perception of being alone and isolated) developed in a significant proportion of the cohort, with reduced social activities significantly mediating this relationship [5]. Furthermore, the findings of another large cross-sectional study found that levels of loneliness were higher when people had less contact with their friends and family [6]. Whether these associations exist in the angling population, which is commonly (although not always) a solitary hobby, is not known. The aim of this study, therefore, was to determine whether social support networks influence feelings of loneliness in a cohort of recreational anglers. 

## 2. Materials and Methods

The full methods of this study have been fully published elsewhere [7]. In brief, participants (anglers and non-anglers) were invited to take part in an online survey from October 2021 to January 2022. The online survey was advertised through Angling Direct and the Tackling Minds Instagram, Facebook and Twitter accounts. Angling Direct also sent the survey link to their mailing list and the link was distributed via the Anglia Ruskin University Twitter account. The survey was open to all UK residents aged 18 years and over. Several variables were collected from participants, including demographic details such as age (years) and gender (Male/Female/Non-binary/Intersex/other), and nine measures of social support. Regarding these social support measures, the following nominal questions were asked to participants: −‘Considering the people you are related through birth, marriage, adoption and so on, how many relatives do you see or hear from at least once a month?’−‘Considering the people you are related through birth, marriage, adoption and so on, how many relatives do you feel at ease with that you can talk about private matters?’−‘Considering the people you are related through birth, marriage, adoption and so on, how many relatives do you feel close to such that you could call on them for help?’−‘How many of your friends do you see or hear from at least once a month?’−‘How many friends do you feel at ease with that you can talk about private matters?’−‘How many friends do you feel close to such that you could call on them for help?’

Questions about loneliness included questions regarding perceived companionship, feeling left out, and feeling isolated. Specifically, these three nominal questions were asked: −‘How often do you feel that you lack companionship?’−‘How often do you feel left out?’−‘How often do you feel isolated from others?’

Participants read a participant information sheet and subsequently provided informed consent prior to the completion of the survey. Ethical approval for the present study was granted by the Anglia Ruskin University Sport and Exercise Science Ethics Panel. 

### Analysis

All statistical analyses were conducted using SPSS Version 28. 0.0.0 (190) (Chicago, IL, USA). Data from the social correlate variables were analyzed using a descriptive analysis. To determine the associations between social correlates (how many family/friends they heard from at least once a month; how many family/friends did they feel at ease with to talk about private matters; how many family/friends did they feel so close to that they could call on them for help; how often did they feel that they lacked companionship; how often did they feel left out; and how often did they feel isolated from others), a Mantel–Haenszel test of trend was performed. 

## 3. Results

Originally, a total of 1792 anglers completed the online survey; however, of this sample there was a very small number of female respondents (*n* = 40). It was therefore deemed appropriate to only include the male sample. In total, 1752 male participants completed the online questionnaire. The majority of participants were either aged between 55 and 64 (25.8%), 45 and 54 (20.8%), or 65 and 74 (20.4%). 

Regarding how many relatives male anglers heard from at least once a month, the majority of participants answered that they either heard from 3 to 4 (34.5%), or 5 to 8 (27.8%) relatives. When the same question was asked about how many friends they heard from at least once a month, the majority of participants had also heard from 3 to 4 (29.6%), or 5 to 8 (18.9%) friends. Regarding how many relatives participants felt at ease with that they could talk about private matters, the majority of participants stated 3–4 (27.5%), followed by 1 (22.1%). When the same question was asked regarding friends, the results were similar, with 27.6% of participants saying that they had 3–4 friends they could talk to about private matters. Regarding how many relatives they felt close to such that they could call on them for help, the majority of participants indicated 3–4 (31.9%), followed by 2 (20.7%). Results were similar when asked the same question about friends that they could call on for help. See Table 1 for all the information. 

Regarding companionship, most participants (57.5%) stated that they hardly ever or never felt a lack of companionship. Regarding feeling left out, 55.2% stated that they hardly ever or never felt left out. Lastly, regarding isolation, the majority of participants (57.2%) stated that they hardly ever or never felt isolated from others. Table 2 shows all the information. 

### 3.1. Trends 

#### 3.1.1. Lack of Companionship

The Mantel–Haenszel test of trend between how many relatives recreational anglers saw or heard from once a month and how often they felt a lack of companionship showed a statistically significant linear association (χ^2^(1) = 85.80, *p* < 0.001, and r = −0.22). A similar trend was also found regarding the association between how many friends recreational anglers saw or heard from once a month and how often they felt a lack of companionship (χ^2^(1) = 92.96, *p* < 0.001, and r = −0.23). A significant trend was also found between how many relatives recreational anglers felt they could talk about private matters with and how often they felt a lack of companionship (χ^2^(1) = 117.40, *p* < 0.001, and r = −0.32), with a similar trend found for friends (χ^2^(1) = 117.40, *p* < 0.001, and r = −0.29). Lastly, a significant trend was also found between how many relatives recreational anglers felt that they could call for help and how often they felt a lack of companionship (χ^2^(1) = 207.52, *p* < 0.001, and r = −0.35), with a similar trend being found for friends (χ^2^(1) = 208.91, *p* < 0.001, and r = −0.35). 

#### 3.1.2. Feeling Left Out

A Mantel–Haenszel test of trend between how many relatives recreational anglers saw or heard from once a month, and how often they felt left out showed a statistically significant linear association (χ^2^(1) = 94.79, *p* < 0.001, and r = −0.23). A similar trend was also found regarding the association between how many friends recreational anglers saw or heard from once a month, and how often they felt left out (χ^2^(1) = 122.73, *p* < 0.001, and r = −0.27). A significant trend was also found between how many relatives recreational anglers felt they could talk about private matters with and how often they felt left out (χ^2^(1) = 197.71, *p* < 0.001, and r = −0.34), with a similar trend found for friends (χ^2^(1) = 184.42, *p* < 0.001, and r = −0.33). Lastly, a significant trend was also found between how many relatives recreational anglers felt that they could call for help and how often they felt left out (χ^2^(1) = 203.51, *p* < 0.001, and r = −0.34), with a similar trend being found for friends (χ^2^(1) = 212.61, *p* < 0.001, and r = −0.35). 

#### 3.1.3. Feelings of Isolation

A Mantel–Haenszel test of trend between how many relatives recreational anglers saw or heard from once a month, and how often they felt isolated showed a statistically significant linear association (χ^2^(1) = 129.48, *p* < 0.001, and r = −0.27). A similar trend was also found regarding the association between how many friends recreational anglers saw or heard from once a month, and how often they felt isolated (χ^2^(1) = 156.73, *p* < 0.001, and r = −0.30). A significant trend was also found between how many relatives recreational anglers felt they could talk about private matters with and how often they felt isolated (χ^2^(1) = 245.46, *p* < 0.001, and r = −0.35), with a similar trend found for friends (χ^2^(1) = 191.83, *p* < 0.001, and r = −0.33). Lastly, a significant trend was also found between how many relatives recreational anglers felt that they could call for help and how often they felt isolated (χ^2^(1) = 241.42, *p* < 0.001, and r = −0.37), with a similar trend being found for friends (χ^2^(1) = 230.39, *p* < 0.001, and r = −0.36).

## 4. Discussion

This study examined trends between social support correlates and feelings of isolation in a cohort of male recreational anglers. The results of this study showed that the higher the number of friends or family that anglers hear from and feel close to, the less likely they are to report a lack of companionship, the less likely they are to report feeling left out, and the less likely they are to report feelings of isolation. These findings are in agreement with previous studies that have found similar associations between social support and loneliness in general populations [6]; however, this is the first study to report these findings in a cohort of recreational anglers. The findings in this specific population are important in that recreational angling is often a solitary hobby/sport; therefore, feelings of loneliness may differ with this population. Our findings suggest this is not the case. Indeed, a significant proportion of this sample reported ‘hardly ever or never’ having feelings of a lack of companionship, feelings of being left out, and feelings of loneliness, indicating that recreational fishing can have positive benefits to overall social wellbeing, or, at the very least, not be detrimental to overall wellbeing. Indeed, higher levels of social support have generally been reported to be associated with greater levels of wellbeing across several types of populations [8,9,10]. A Turkish study of recreational anglers found that 65.2% of people angled recreationally to be with friends, with 22.6% having done so to be alone. Furthermore, they found that a key motivation for recreational anglers was to ‘be happy’ [11]. This, combined with the results of this study, add to the growing body of evidence for the potential positive benefits of recreational angling. Further studies should aim to compare anglers with non-anglers to determine if differences exist between the two populations. 

Findings from this study should be interpreted in light of its limitations. Firstly, because of the cross-sectional nature of the study, and the impossibility of determining dependent and independent variables in the Mantel–Haenszel test, causal inferences could not be made. Future longitudinal cohort studies are now required. Secondly, the methods of recruiting the sample of recreational anglers could be open to potential selection bias. For example, the use of an online survey could exclude groups who are less likely to have internet access or access to social media. Therefore, populations with lower incomes, with lower levels of computer literacy, and older adults may be under-represented in this study. Thirdly, the study variables were self-reported and thus potentially introduced social desirability and recall bias into the study findings. Finally, very few females responded to the online survey and thus were removed from the present analysis. Future research in the female angling population is now required. 

## 5. Conclusions

In conclusion, in the present study including a sample of 1752 male recreational anglers from the UK, it was found that those who reported having more friends and family that they hear from, can call upon for support, and trust enough to talk to about private matters are less likely to report feelings of a lack of companionship, feelings of being left out, and feelings of isolation. Furthermore, a significant proportion of recreational anglers reported having very few feelings of lack of companionship, feelings of being left out, and feelings of isolation, despite having a sometimes-solitary hobby. Future longitudinal cohort studies are now required before any concrete recommendations can be provided. Moreover, it would be prudent to repeat the present study in a representative sample of female anglers to determine whether the findings also apply to females. 

## Figures and Tables

**Table 1 ijerph-20-05997-t001:** Distributions of social support questions in male anglers.

	0	1	2	3–4	5–8	9 or More
How many relatives do you hear from at least once a month?	3.7% (65)	7.7% (134)	16.2% (283)	34.5% (602)	27.8% (485)	10.1% (177)
How many of your friends do you see or hear from at least once a month?	9.6% (167)	11.4% (199)	16.4% (286)	29.6% (516)	18.9% (329)	14.1% (246)
How many relatives do you feel at ease with that you can talk about private matters?	15.6% (272)	22.1% (385)	20.8% (363)	27.5% (480)	10.3% (179)	3.8% (67)
How many friends do you feel at ease with that you can talk about private matters?	20.1% (350)	17.8% (310)	22.8% (398)	27.6% (481)	7.6% (133)	4.0% (70)
How many relatives do you feel close to such that you could call on them for help?	9.2% (161)	15.3% (266)	20.7% (361)	31.9% (557)	16.5% (287)	6.4% (112)
How many friends do you feel close to such that you could call on them for help?	15.3% (266)	16.5% (287)	23.9 (416)	28.4% (495)	9.5% (166)	6.4% (111)

**Table 2 ijerph-20-05997-t002:** Distributions of companionship questions in male anglers.

	Hardly Ever or Never	Some of the Time	Often
How often do you feel that you lack companionship?	57.5% (1004)	31.3% (546)	11.2% (196)
How often do you feel left out?	55.2% (962)	34.5% (601)	10.3% (179)
How often do you feel isolated from others?	57.2% (996)	31.1% (541)	11.8% (205)

## Data Availability

All data for this study are available from the corresponding authors upon reasonable request.

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
