# Peer review of "The Relationship between Social Support Correlates and Feelings of Loneliness among Male UK Recreational Anglers"

_ijerph, 2023, doi:10.3390/ijerph20115997_

Round 1

Reviewer 1 Report

Thank you for the opportunity to review this interesting manuscript. The authors have investigated social correlates between anglers ad non-anglers, based on previous evidence on the benefit of blue spaces. The results are interesting – albeit not unexpected, with anglers having less social support.

Issue – sampling bias. Anglers were accessed through suitable channels that they access. However, accessing non-anglers through social media might create a bias as those who are more social and have stronger/more frequent with others. Additionally, the overwhelming majority of anglers in the sample (94% vs. 6%) puts the reliability of the comparisons into question.

Additionally, the nature of angling is such that it conducive to extended periods of time alone. Therefore, people who are naturally more introverted might be more likely to choose this activity. This is also demonstrated by the (perceived) lack of difference for the lack of companionship, being left out, and feeling isolated, as the authors discuss on l. 97-99.

Therefore, I think that the authors’ statement that anglers may be getting involved in angling to fill a social void (l.102-105).

To provide a better comparison, the authors need to obtain responses from a larger number of non-anglers recruited using suitable channels. Additionally, including questions of the reasons for angling/non-angling and the perceived outcomes for the person would provide more substance to the study.

Some minor specific issues:

L. 22 ‘one’ a month – should be ‘once’ a month

L.43 – ‘is’ to examine – was to examine (past tense for reporting completed studies)

L. 59-60 and L61-62 – repeated information, remove one or check if one of these should be ‘close’ rather than ‘ease’

Author Response

Comment

Response

Issue – sampling bias. Anglers were accessed through suitable channels that they access. However, accessing non-anglers through social media might create a bias as those who are more social and have stronger/more frequent with others. Additionally, the overwhelming majority of anglers in the sample (94% vs. 6%) puts the reliability of the comparisons into question.

Thanks for this comment. Note that we have addressed this by only including anglers and males in the new sample and anal

Additionally, the nature of angling is such that it conducive to extended periods of time alone. Therefore, people who are naturally more introverted might be more likely to choose this activity. This is also demonstrated by the (perceived) lack of difference for the lack of companionship, being left out, and feeling isolated, as the authors discuss on l. 97-99.

Therefore, I think that the authors’ statement that anglers may be getting involved in angling to fill a social void (l.102-105).

Thanks for these comments. Because of the nature of the sample, we have decided to change the analyses and sample.

We are now measuring whether having social support from family or friends influences levels of companionship, feelings of being left out, and feelings of isolation in this unique population.

To provide a better comparison, the authors need to obtain responses from a larger number of non-anglers recruited using suitable channels. Additionally, including questions of the reasons for angling/non-angling and the perceived outcomes for the person would provide more substance to the study.

L. 22 ‘one’ a month – should be ‘once’ a month

All incidences of these have now been changed.

L.43 – ‘is’ to examine – was to examine (past tense for reporting completed studies)

This has now been changed.

L. 59-60 and L61-62 – repeated information, remove one or check if one of these should be ‘close’ rather than ‘ease’

This has now been changed.

Reviewer 2 Report

Overall, a good report. Here are a few minor corrections to consider:

Some minor grammar corrections: In line 22 you probably mean once a month not "one a month".

In lines 38-40 could you be more specific on the precise levels of loneliness you mean? High levels, low levels, various levels?

Line 41: You probably mean "are the differences" or "is the difference" not "is the differences".

It looks like there is a huge disparity between the number of anglers (94%) and non-anglers (6%) in your study. This could have skewed your findings in one direction? Could you explain your reasons for this?

Author Response

Comment

Response

Some minor grammar corrections: In line 22 you probably mean once a month not "one a month".

This has now been changed.

In lines 38-40 could you be more specific on the precise levels of loneliness you mean? High levels, low levels, various levels?

This has now been changed to ‘high levels of loneliness’

Line 41: You probably mean "are the differences" or "is the difference" not "is the differences".

This has now been changed. Thank you for this.

It looks like there is a huge disparity between the number of anglers (94%) and non-anglers (6%) in your study. This could have skewed your findings in one direction? Could you explain your reasons for this?

This is a completely relevant point – we have changed the sample to include only anglers and men now.

Round 2

Reviewer 1 Report

Thank you for revising the paper, it is much improved!